# Micropropagation of Mountain Mulberry (*Morus bombycis* Koidz.) ‘Kenmochi’ on Cytokinin-Free Medium

**DOI:** 10.3390/plants9111533

**Published:** 2020-11-10

**Authors:** Wojciech Litwińczuk, Beata Jacek

**Affiliations:** Department of Plant Physiology and Biotechnology, Institute of Agricultural Sciences, Land Management and Environmental Protection, University of Rzeszów, Ćwiklińskiej 2nd St., 35-601 Rzeszów, Poland; jacekbeat@interia.pl

**Keywords:** single-node culture, axillary branching, in vitro

## Abstract

The aim of the study was to compare two methods of micropropagation of mulberry: single-node culture (“SNC”), and axillary-branching (“AxB”). The experiments were carried out on in vitro cultures for 6 successive passages. The “AxB” cultures were propagated on modified MS medium (+ 25% Ca^2+^ and Mg^2+^), supplemented with WPM vitamins, sucrose (30 g L^−1^), and BA (1.5 mg l^-1^). The “SNC” cultures were grown on cytokinin-free 1/2 MS (macro- and micronutrients) medium supplemented with WPM vitamins, IBA (0.05 mg l^-1^), and sucrose (15 g l^-1^). Both media (pH 5.8) were solidified with agar (7.0 g l^-1^). Initiation of in vitro cultures from explants taken from adult trees and young, potted plants was feasible on both media. Cultures were established from about 1 cm long nodal explants. Generally “SNC” cultures formed one well rooted, significantly longer axillary shoot with bigger leaves than “AxB” cultures, which developed significantly more shoots and big callus at the explant base. All shoots collected from “SNC” and “AxB” cultures rooted in vivo in peat mixture and developed into similar plantlets. The single-node method based on application of cytokinin-free medium is a good alternative for the axillary-branching method for micropropagation of mountain mulberry (*Morus bombycis*) ‘Kenmochi”’.

## 1. Introduction

Mulberries (*Morus* sp.) are valuable trees or shrubs which can be used in many ways (sericultural and pharmaceutical industry, ornamental tree, edible fruit, valuable timber, biomass) [1,2,3]. They are host plants for mulberry silkworm (*Bombyx mori*). Mulberry-derived products are used in traditional medicine against diabetes, throat inflammations, dysentery, constipation, and helminthiasis [2,3,4,5]. It seems that next to valuable clones of white mulberry (Morus alba L.) the Japanese cultivar ‘Kenmochi”’ of mountain mulberry (*Morus bombycis* Koidz.) deserves attention. It is designed for sericulture and is intensively cultivated in cold areas of Japan [6]. It was confirmed by Alipanah et al. [7] who showed that silkworms that consumed leaves of ‘Kenmochi”’ had better performance than silkworms fed leaves of white (*Morus alba* L.), and black mulberry (*Morus nigra* L.) clones. The quality of cocoons was improved as well. However, cultivar ‘Kenmochi”’ may be also considered to be an ornamental tree. It is sufficiently winter-hardy in Poland. It bears edible black/dark red slightly sour and sweet fruit. They taste better than bland white mulberry fruit. The fruit are liked by wild birds. It is worth mentioning that ‘Kenmochi”’ fruit are characterized by high phenolics content and strong radical scavenging activity and reducing power [8]. For these reasons, cultivar ‘Kenmochi”’ might be useful in growing orchards, edible landscaping, and agroforestry in Poland. Conventional propagation of mulberries (cuttings, grafting, etc.) is difficult and troublesome, especially in countries with a moderate climate [1,9,10,11]. Therefore, the main way of mass cloning of valuable mulberry cultivars is micropropagation. Plants of many species are propagated in vitro through somatic embryos or organ cultures. In the last case, the method based on stimulation of branching of shoots, preferably axillary ones, is most often used. Such ways are called “Axillary Branching” (AxB) or “Multi-Apexed Shoots” depending on propagation of species with long or short internodes, respectively. To promote proliferation of axillary shoots (and obtain a high multiplication factor) the application of cytokinins is essential. However, in such conditions the adventitious shoots may also spontaneously develop from leaves, internodes or callus [12,13,14,15,16,17]. They are often difficult to distinguish and separate from axillary ones [14,15]. Adventitious shoots which develop in vitro in cultures of many species are suspected to be the main source of somaclonal variation [18]. In summary, “AxB” method is very efficient thus is used in mass propagation of elite, healthy plants. However, the risk of obtaining plants with a changed phenotype should be considered as well. The other form of micropropagation is “Single-Node Culture” (SNC) [19]. It is based on the application of media devoid of cytokinins, sometimes supplemented with low doses of auxins to stimulate rhizogenesis. In such conditions, the initial explant usually forms one strong shoot from the existing axillary bud. It is usually divided into stem fragments with one (single) node to establish the next subculture. The development of adventitious shoots is not observed. Therefore, the main advantage of the “SNC” method is the lowest risk of somaclonal variation, whereas the main inconvenience—usually—is the low propagation rate. Its use is limited only for plants with long internodes and strong apical dominance. Therefore, it is seldom used in micropropagation of different species [19]. Many studies have been carried out on in vitro cultures of mulberries (*Morus* sp.) since 1970 when the first article on this topic was published [20] and some, but similar, protocols of successful micropropagation through axillary branching was described [among others: [1,3,5,9,10,11,21,22,23,24,25,26,27,28,29,30,31,32]]. However, some problems related to mulberry micropropagation still exist. The cultures grown on media supplemented with cytokinins usually form a big callus at the explant base [5,21,22,24,25]. It is especially visible in the case of mountain mulberry ‘Kenmochi”’ in vitro cultures where the callus develops even from lenticels (Figure 1). It decreases shoot proliferation and worsens culture quality and health. It seems that mulberry in vitro cultures synthesize endogenous auxins as shoot cultures spontaneously roots during proliferation stage and the addition of auxins to the media stimulates both callusing and rhizogenesis. The micropropagated shoots also easily root both in vitro and in vivo [23]. As an apical dominance is observed, it is worth checking whether mulberries could be propagated in vitro according to single-node culture (“SNC”) method in a medium devoid of cytokinins. This was the aim of the present study. 

## 2. Materials and Methods

The experiments were carried out on in vitro cultures or plants of mountain mulberry (*Morus bombycis* Koidz.) ‘Kenmochi”’. First plants were kindly provided from the Institute of Pomology and Floriculture in Skierniewice (Poland) in 1990. Their vegetative progeny has been grown on “Zalesie” Campus of University of Rzeszów, Poland since 1995. Mulberry plants were micropropagated on a small scale in the university plant nursery. In the present study, the control cultures were multiplied through commonly used, standard, axillary-branching (“AxB”) method based on the application of cytokinins in initiation and multiplication stages. The modified MS [33] medium with elevated Ca^2+^ and Mg^2+^ dose, supplemented with 6-benzylaminopurine (BA) was used for this purpose. Alternatively, the tested cultures were propagated through single-node (“SNC”) method on cytokinin-free, double-diluted MS medium which favors rhizogenesis [23]. However, it was also used in the initiation and multiplication stages. Compositions of studied media are presented in Table 1. Cultures were incubated at 25 ± 1.5 °C and photoperiod of 16 h/8 h (day/night) at approximately 26 μmol·m^−2^·s^−1^ PPFD, provided by cool white fluorescent lamps (OSRAM). Similar temperature and light conditions were used in many procedures of mulberry micropropagation [3,23,24,26,27,28,29,30].

### 2.1. Initiation of In Vitro Cultures

In the case of culture initiation, the initial (single-node) explants were prepared from new spring shoots collected from adult, at least 20-year-old trees and from one-year-old nursery plants in April. The experiments were carried out on 30 and 40 explants, respectively. Before initiation, the shoots were soaked in fungicide “Funaben T” (carbendazim 20% and thiram 45%) 0.1% solution for 12 h. Then the pieces of shoots were dipped in ethanol 70% for a few seconds and sterilized in commercial “ACE lemon” 10% solution in ultrasonic bath for 15 min. After this, they were roughly rinsed with autoclaved deionized water and finally in sterile solution of citric acid and vitamin C (50 mg L^−1^, both). The single-node explants were put in vials filled with the same media (“AxB”, “SNC”) as used in the next experiments. However, they were additionally supplemented with PPM^®^ (1 mL l^−1^). After four weeks, the whole, healthy-looking cultures were placed in jars on (“AxB”, “SNC”) media devoid of PPM^®^. They were grown in the same light/temperature conditions as in multiplication stage. At the end of passage, the number of growing cultures were recorded.

### 2.2. Multiplication Stage (Main Experiments)

The main experiments concerning the multiplication stage were carried out on well-established, about 2-year-old in vitro cultures. Usually, about 1-cm-long nodal explants were used. Generally, the shoots grown in the presence of BA had distinctly short internodes than obtained in BA-free medium (Figure 2). Therefore the explants prepared from control (“AxB”) cultures consisted of 2–3 nodes whereas those procured from “SNC” cultures comprised of one node (Figure 3). In additional experiments, other types of explants were also tested (longer, 2-cm-long two-node pieces of shoots or shoot tips 1 cm long). 

Cultures were grown in vitro in glass jars (450 mL) with ventilated polypropylene twist lids, filled with medium (60 mL). Usually, each treatment (“AxB” and “SNC”) in each subculture was represented by 4 jars (×8 explants). The exceptions were: the first subculture (3 jars × 8 explants) and the sixth one (5 jars ×8 explants). The experiments were carried out through six subsequent 6–7-week-long subcultures. In order to check whether “the switch” from “SNC” to “AxB” method is possible, the explants prepared from “SNC” cultures were grown on control “AxB” medium. Such additional treatment was also represented by 4 jars (×8 explants) in three subcultures, and 5 jars (×8 explants) in the sixth subculture. At the end of passage, the number and length of shoots and callus size was determined. The number of growing cultures and rooted explants were also recorded.

### 2.3. Ex Vitro Rooting of Shoots (Additional Experiment)

Shoots (at least 2 cm long with shoot tip) were collected from “AxB” and “SNC” cultures. They were rooted in vivo without application of auxins and fungicides in commercial peat mixture Bella^®^ (Hydrokomplet) designed for ornamental pot plants in mist chambers (near 100% RH) with transparent plastic covers. The same temperature and light conditions were set as during in vitro propagation. The covers were gradually disclosed from the first week and removed after the third week. Then after the next week the number and size of obtained plantlets were recorded. The leaf greenness index (relative chlorophyll content) expressed in SPAD units was determined for two young but fully developed leaves of each plant using a portable KONICA MINOLTA Chlorophyll Meter SPAD-502Plus. Forty “AxB” and 60 “SNC” shoots were used in such experiment. 

### 2.4. Statistical Analyses

Collected data were subjected to an ANOVA and LSD_0.05_ mean separation test using Statistica 9.0 computer software. Data presented as percentages (number of growing cultures, number of rooted shoots in vitro, number of obtained plantlets in vivo) were subjected to testing on difference between two proportions. The differences were considered to be significant at α = 0.05. 

## 3. Results 

### 3.1. Initiation of In Vitro Cultures 

It was found that initiation of in vitro mulberry cultures from initial explants taken from adult trees and young, potted plants was feasible both on standard, cytokinin-containing MS (“AxB”) and on cytokinin-free ½ MS (“SNC”) media (Table 2, Figure 4). Initiation in cytokinin-supplemented medium was much more effective than on cytokinin-free ones, while initial explants were prepared from a field tree. Usually, together with growing shoots, the initial cultures formed calluses in the presence of BA whereas roots—in its absence (Figure 4a–d, respectively). However, some new shoots which grown from initial explant on "SNC” medium formed dormant buds and completed growth. The differences between treatments were not so distinct while initial explants were taken from nursery plants (Table 2). 

### 3.2. Multiplication Stage (Main Experiments)

Similar numbers of explants placed on PGRs-free (“SNC”) and control (“AxB”) media started to grow in all 6 subcultures of mountain mulberry ‘Kenmochi”’ (Table 3). All cultures, both “AxB” and “SNC” ones, consisted of axillary shoots (without adventitious ones) at the end of subculture. 

The single-node “SNC” explants rooted easily, and the callusing of the explant base was not observed (Table 3, Figure 5). As a rule, the “SNC” explant put on BA-free medium developed one long axillary shoot with big leaves (Figure 3). In comparison, a nodal explant placed on BA-containing media developed big callus (sometimes with few roots) and significantly more shoots (Table 3, Figure 5). However, such shoots were distinctly shorter in almost all subcultures. It was due to a significantly lower number of nodes (4.3 < *s* 5.5) and shorter internodes (1.0 cm < 1.4 cm) of obtained shoots (“AxB” *vs* “SNC” cultures respectively, the results for the first passage). A clear relationship between the micropropagation method and the total length of shoots formed by cultures was not found (Table 3).

The old “SNC” cultures were able to develop new axillary shoots after the harvest of the first ones (Figure 6). However, new shoots were significantly shorter than those collected previously (Table 4). On the contrary, the “AxB” cultures usually did not form new shoots after harvesting the old ones. 

The cultures obtained from explants prepared from “SNC” cultures and placed in medium supplemented with BA developed bigger callus and less axillary shoots than control ones (Table 5). The clear relationship between the origin of explants and elongation of shoots was not found. The cultures established from single-node and two-node explants grown in rooting (“SNC”) medium resembled each other (Table 6). On the other hand the growth of cultures on multiplication (“AxB”) medium was strongly dependent on explant type as shoot-tip explants developed significantly more and longer axillary shoots than nodal explants (Table 7). 

### 3.3. Ex Vitro Rooting of Shoots (Additional Experiment)

All shoots collected from “SNC” and “AxB” cultures rooted in vivo. The fully acclimated plantlets reached similar quality (Table 8, Figure 7). Any differences in obtained plantlet height, size of leaf blade, or relative chlorophyll content were not found. 

## 4. Discussion

The aim of the present study was to assess the possibility of micropropagation of mountain mulberry ‘Kenmochi”’ through single-node culture method and compare it with the standard, axillary-branching method based on application cytokinin-containing medium. Such a study was undertaken as during the standard “AxB” method; some problems, such as excessive growth of calluses resulting in the deterioration of shoot proliferation and quality, were encountered (Figure 1) [5,21,22,24,25]. It should be underlined that the establishment of strict experiments was neither possible nor recommended because of the different morphology of shoots obtained through the compared methods and difficulty of the application and preparation of the same explants. For example, 1-cm-long explants for the “SNC” method usually consisted of one node, long internode, and trimmed or removed leaf blade, whereas for “AxB” method the same was from 2–3 nodes and at least one uncut leaf (Figure 3).

The single-node culture method is known but seldom used in micropropagation of different plants [19]. The main advantage of it is the lowest risk of somaclonal variation, whereas the main inconvenience is usually low propagation rate. The spontaneous development of adventitious shoots was rather unobserved in the present study, because overgrown callus slightly complicated identification of shoot origin. However, such a phenomenon was not notified by other authors, who propagated mulberries through the routine “AxB” method, as well. Of course, direct or indirect organogenesis of adventitious shoots might be induced in in vitro cultures of mulberry [5]. However, the initial explants must be specially pretreated and placed on modified media, supplemented among others with TDZ or higher doses of BA [5,35,36,37,38,39]. Since 1970, when the first article about successful micropropagation of mulberry was published [20], only one description of somaclonal variant has appeared [35]. Nevertheless, it was deliberately obtained through indirect regeneration (from callus). Therefore, it seems that the risk of somaclonal variation is low both in the case use of “AxB” and “SNC” methods. Nevertheless, it cannot be completely excluded, since to date the results of molecular analyses on genetic or epigenetic status of mulberry plants propagated through routine “AxB” method, including the present study, have not been published. Contrary to the common opinion, it seems that in the case of mulberries, the “SNC” method is more productive than the “AxB” one, despite the worsening shoot proliferation (Table 3). Generally, for mulberry micropropagation, both shoot tips and nodal segments could be exploited [30,31,33,40]. Nevertheless, for the “AxB” method, the shoot tip is the most recommended explant type, and the usage of other types of explants gives worse results, such as slower shoot growth in the case of mountain mulberry (Table 7). Such a phenomenon was confirmed also in previous experiments carried out on white mulberry clones [23]. Thus, the main indicator of micropropagation efficiency is the number of axillary shoots (shoot tips) in the case of “AxB” method contrary to “SNC” method. In such a case, the proper indicator is the length of shoot or the number of phytomers, as single-node explants develop into a healthy culture/plant in vitro. In a supplementary experiment (12th subculture) we repeated additional measurements, as in the first subculture. The obtained results were similar. We found that “AxB” cultures produced, as usual, significantly more shoots than “SNC” ones (1.6 > 1.0) but obtained shoots consisting of similar number of nodes (5.4 *vs.* 5.5, ns, respectively). Nonetheless, the mean length of shoots and internodes was significantly higher in the case of “SNC” shoots than “AxB” shoots (6.5 cm > 2.3 cm and 1.2 cm > 0.4 cm, respectively (Figure 2)). Thus, the differences in shoot/internode elongation were more distinct than observed in the first subculture (Table 3). Therefore from one “SNC” culture about 5–6 nodal explants could be prepared, whereas about 3–4 explants (2 shoot tips and 1–2 nodal explants) could be prepared from “AxB” single culture. In addition, the second harvest of “SNC” shoots from the same culture vessel may also be obtained (Table 4, Figure 6). On the other hand it should be pointed that “SNC” cultures form shoots with big leaves should be cut off or trimmed during explant preparation, which may slightly increase the labor intensity. If need be, the used medium should be modified to reduce the size of leaf blades. It is possible to use explants obtained from “SNC” cultures in “AxB” method but it is not recommended as obtained cultures develop bigger calluses and fewer axillary shoots (Table 5). The shoots obtained both from “SNC” and “AxB” cultures, rooted easily in vivo and regenerated plantlets were of the same quality (Figure 7, Table 8). It is worth mentioning that “SNC” shoots are much longer than the “AxB” shoots, thus are easier to manipulate (Table 3). They could also be divided into 2 pieces and rooted in vivo separately with similar success. Thus, “SNC” cultures supplied even more shoots suitable for rooting than “AxB” ones 

## 5. Conclusions

In summary, the “SNC” method provides at least similar propagation efficiency and quality of obtained cultures and plantlets. It is also cheaper, as the medium without cytokinins, with lowered concentration of sucrose and macronutrients, is used. Thus, the single-node method based on application of cytokinin-free medium is a good alternative for the axillary-branching method for micropropagation and/or long-term in vitro storage of mountain mulberry (*Morus bombycis* Koidz.) ‘Kenmochi”’. The main properties, advantages, and disadvantages of two methods of micropropagation are given in Table 9. It is worth mentioning that similar observations were made in the case of two white mulberry (*Morus alba* L.) clones (Figure 8). 

## Figures and Tables

**Figure 1 plants-09-01533-f001:**
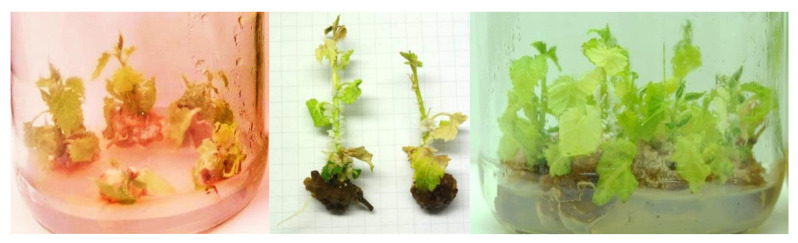
Abnormal development of in vitro shoot cultures of mountain mulberry ‘Kenmochi”’ grown on medium supplemented with cytokinin (BA). Note the excessive growth of a callus connected with suppressed shoot proliferation, growth of calli-like tissues from lenticels, and apical shoot necrosis.

**Figure 2 plants-09-01533-f002:**
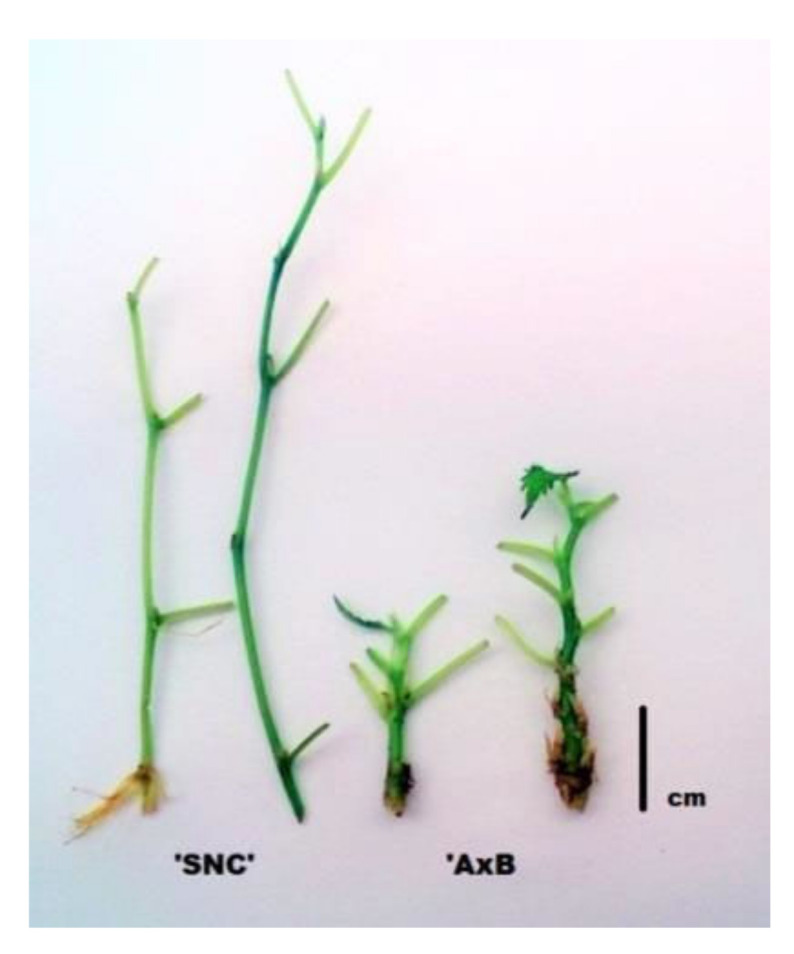
Morphological differences of shoots obtained from “SNC” and “AxB” cultures of mountain mulberry ‘Kenmochi”’. The bar is 1 cm long.

**Figure 3 plants-09-01533-f003:**
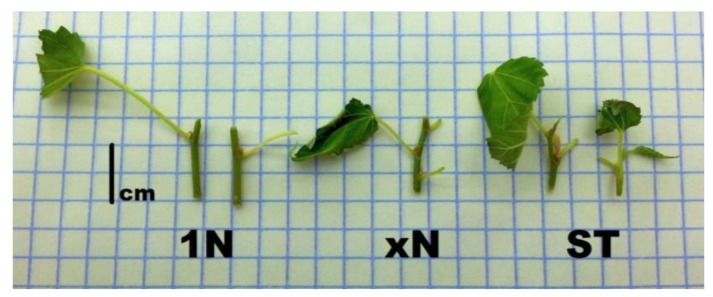
Types of explants used in experiments. 1N—single-node explants prepared from “SNC” cultures; xN—several node explant prepared from “AxB” cultures; ST—shoot-tip explants obtained from “AxB” or “SNC” cultures. The bar is 1 cm long.

**Figure 4 plants-09-01533-f004:**
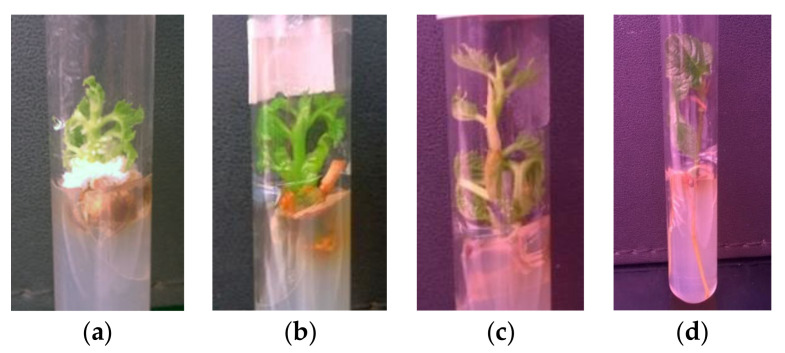
Initiation of in vitro shoot cultures of mountain mulberry ‘Kenmochi”’ on cytokinin-containing (“AxB”) and cytokinin-free (“SNC”) media (**a**–**d**, respectively). Note growth of callus on “AxB” medium and emergence of roots on “SNC” medium.

**Figure 5 plants-09-01533-f005:**
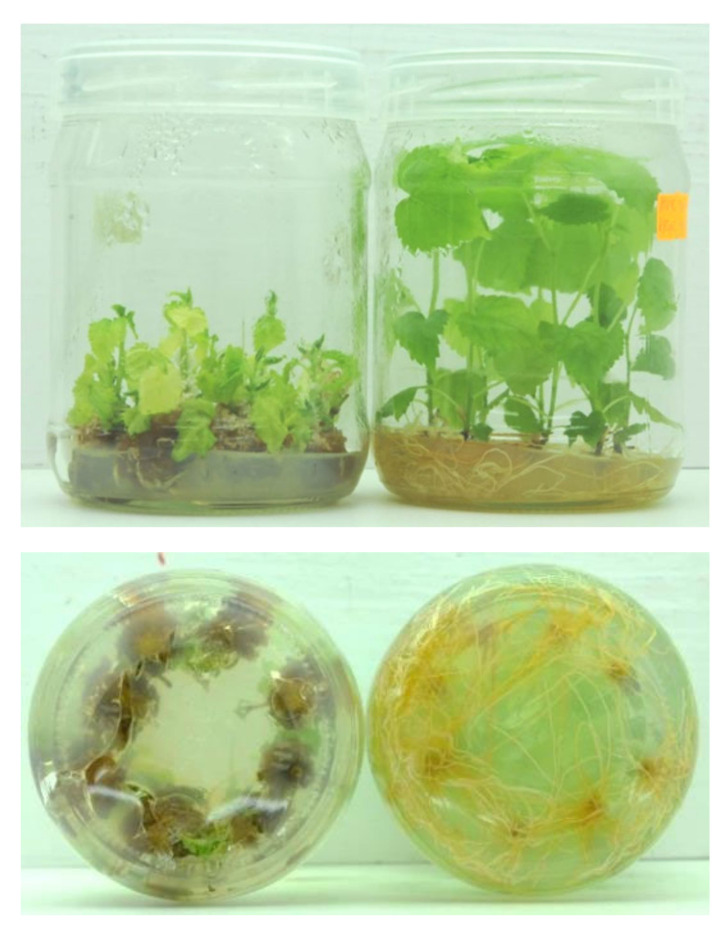
Performance of in vitro shoot cultures of mountain mulberry ‘Kenmochi”’ propagated through axillary-branching (left jar) and single-node (right jar) method (on cytokinin-containing and cytokinin-free media, respectively). Note the differences in elongation of shoots, immoderate growth of callus in “AxB” cultures, and profuse growth of roots in “SNC” cultures.

**Figure 6 plants-09-01533-f006:**
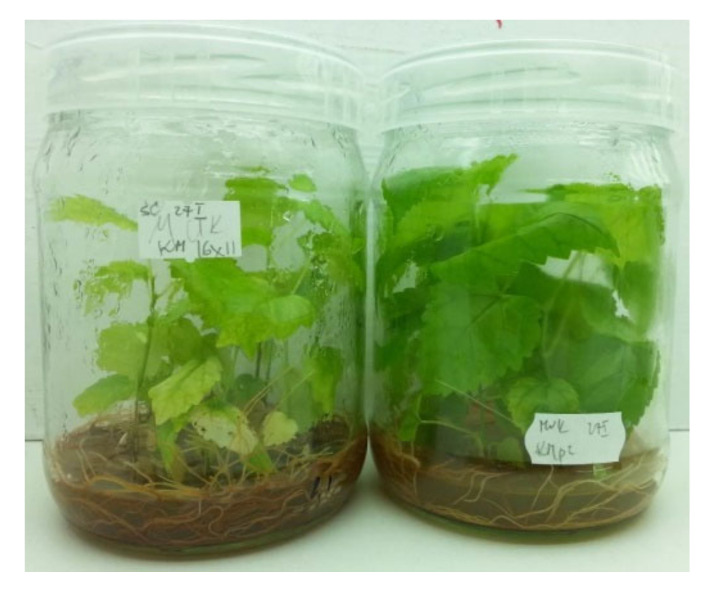
Comparison of the regrowth of axillary shoots in old cultures after the first harvest of shoots (left jar) with newly established “SNC” subculture (right jar). Both cultures of mountain mulberry ‘Kenmochi”’ were grown on cytokinin-free media. Old cultures after 12 weeks of growth and 6 weeks after first harvest. New cultures after 6 weeks of growth.

**Figure 7 plants-09-01533-f007:**
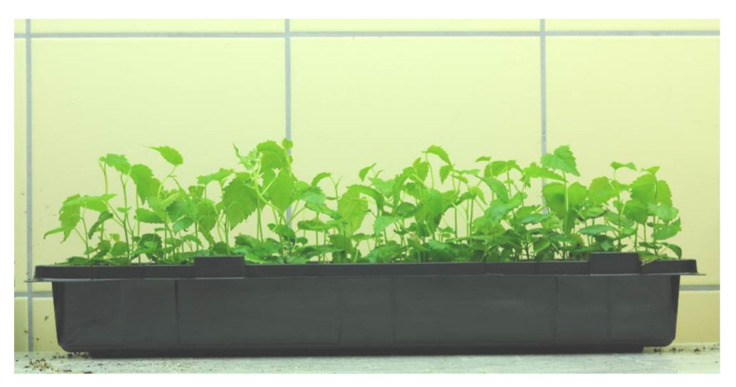
Fully acclimated plantlets of mountain mulberry ‘Kenmochi”’ obtained by rooting microshoots in vivo four weeks ago.

**Figure 8 plants-09-01533-f008:**
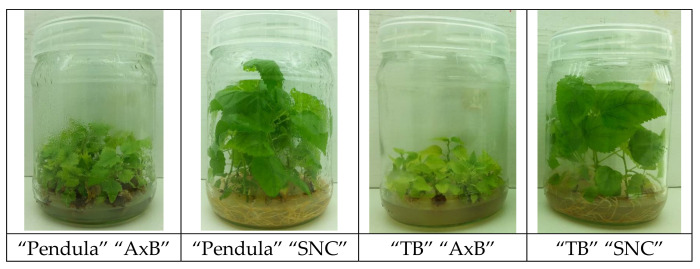
Performance of in vitro shoot cultures of white mulberry clones grown on cytokinin-containing (“AxB”) and cytokinin-free (“SNC”) medium. “TB”—”Tarnobrzeska”—local clone.

**Table 1 plants-09-01533-t001:** Composition of media used for micropropagation of mountain mulberry.

Media Constituents	“AxB” Medium (Standard)	“SNC” Medium
Macronutrients		50% MS ^1^
N salts	100% MS	
K salts	100% MS	
P salts	100% MS	
Ca salts	125% MS	
Mg salts	125% MS	
Micronutrients	100% MS	50% MS
Vitamins	100% WPM^2^	100% WPM
*Myo*-inositol	100 mg L^−1^	100 mg L^−1^
Sucrose	30 g L^−1^	15 g L^−1^
BA	1.5 mg L^-1^	-
IBA	-	0.05 mg L^−1^
Agar	7 g L^−1^	7 g L^−1^
pH	5.8	5.8

^1)^ MS [33] macronutrients: NH_4_NO_3_ 1650 mg L^−1^, KNO_3_ 1900 mg L^−1^, CaCl_2_ × 2 H_2_O 440 mg L^−1^, MgSO_4_ × 7 H_2_O mg L^−1^, KH_2_PO_4_ 170 mg L^−1^; MS micronutrients: NaFeEDTA 36.7 mg L^−1^, KI 0.83 mg L^−1^, H_3_BO_3_ 6.3 mg L^−1^, MnSO_4_ × 4 H_2_O 22.3 mg L^−1^, ZnSO_4_ × 7H_2_O 8.6 mg L^−1^, Na_2_MoO_4_ × 2H_2_O 0.25 mg L^−1^, CuSO_4_ × 5H_2_O 0.025 mg L^−1^, CoCl_2_ × 6H_2_O 0.025 mg L^−1^; ^2)^ WPM [34] vitamins: glycine 2.0 mg L^−1^, thiamine HCl 1.0 mg L^−1^, pyridoxine HCl 0.5 mg L^−1^, nicotinic acid 0.5 mg L^−1^. The used media were prepared from self-made stock solutions. All macro- and micronutrient salts (except for Fe) and sucrose (all reagents of “pure p.a.” grade) were provided by Chempur Company whereas NAFeEDTA, aminoacids, vitamins, and PGRs (all chemicals of “plant cell culture tested” grade) by Sigma-Aldrich. The agar (Lab-Agar AB04) was supplied by Biocorp Poland S.A.

**Table 2 plants-09-01533-t002:** Initiation of mountain mulberry ‘Kenmochi”’ in vitro cultures on standard (AxB) and cytokinin-free (SNC) media.

Source of Explants	No of Survived Explants (S) [%]		No of Explants which Started to Grow (G) [%]		Efficiency of Initiation (S x G) [%]	
	“SNC”	“AxB”	“SNC”	“AxB”	“SNC”	“AxB”
field tree	52 a,A	100 b,B	38 a,A	100 b,B	21 a,A	100 b,B
nursery plants	100 b,B	92 a,A	64 a,B	82 a,A	64 a,B	75 a,A

The means marked with various letter are significantly different at α = 0.05. Lowercase letters—difference between treatments (“SNC”/”AxB”); capital letters—difference between explant origin (field/nursery plants); Contaminated cultures were excluded from calculations.

**Table 3 plants-09-01533-t003:** Growth of mountain mulberry ‘Kenmochi”’ in vitro cultures on standard (AxB) and cytokinin-free (SNC) media.

Medium/ No of Subculture	Explant Type	No of Growing Cultures [%]	Number of Shoots [pcs]	Length of the Longest Shoot [cm]	Total Length of Shoots [cm]	Callus Size [cm]
SNC/1	ST	91.7 a	1.0 a	5.9 b	6.0 a	0.0 a
AxB/1	ST	91.7 a	1.8 b*	3.2 a	4.9 a	2.3 b
SNC/2	1N	96.9 a	1.0 a	7.4 b	7.5 b	0.0 a
AxB/2	xN	100 a	1.6 b	2.0 a	2.6 a	1.6 b
SNC/3	1N	100 a	1.0 a	5.2 a	5.2 a	0.0 a
AxB/3	xN	100 a	1.7 b	5.0 a	7.0 b	1.1 b
SNC/4	1N	86.3 a	1.0 a	5.9 b	6.1 b	0.0 a
AxB/4	xN	100 a	1.6 b	1.9 a	2.6 a	1.5 b
SNC/5	1N	100 a	1.0 a	5.5 b	5.5 a	0.0 a
AxB/5	xN	95.6 a	2.1 b	3.2 a	5.3 a	1.7 b
SNC/6	1N	97.5 a	1.0 a	2.8 b	2.8 a	0.0 a
AxB/6	xN	91.5 a	1.4 b	1.9 a	2.5 a	0.9 b
SNC (mean)		95.4 -	1.0 -	5.5 -	5.5 -	0.0 -
AxB (mean)		96.5 -	1.7 -	2.9 -	4.2 -	1.5 -

ST—shoot-tip explants obtained from “AxB” or “SNC” cultures, 1N—single-node prepared from “SNC” cultures, xN—several node explants prepared from “AxB” cultures; “SNC” medium—cytokinin-free medium (rooting medium) designed for single-node culture method of micropropagation; “AxB” medium—cytokinin-supplemented medium (multiplication medium) designed for axillary-branching method of micropropagation; The means marked with various letter are significantly different at α=0.05. Hyphens mean that statistical analysis (for data from 6 subcultures together) was not done because of different number of analysed cultures in the first, sixth and other subcultures (M&M).

**Table 4 plants-09-01533-t004:** Regrowth of axillary shoots of mountain mulberry ‘Kenmochi”’ in vitro cultures on cytokinin-free (SNC) medium.

Type of Culture	No of Growing Cultures [%]	Number of Shoots [pcs]	Length of the Longest Shoot [cm]	Total Length of Shoots [cm]	Callus Size [cm]
Old cultures after one harvest of shoots	86.3 a	1.3 a	2.2 a	2.7 a	0.0 a
New culture	100 a	1.0 a	5.5 b	5.5 b	0.0 a

The means marked with various letter are significantly different at α = 0.05. The observations were made on 4 jars (with 8 cultures) for each treatment.

**Table 5 plants-09-01533-t005:** Growth of mountain mulberry ‘Kenmochi”’ in vitro cultures on standard (“AxB”) medium established from explants prepared from “AxB” and “SNC” cultures.

Origin of Explant/ No of Subculture	Explant Type	No of Growing Cultures [%]	Number of Shoots [pcs]	Length of the Longest Shoot [cm]	Total Length of Shoots [cm]	Callus Size [cm]
from SNC/2 *	1N **	93.8 a	1.3 a	2.7 b	3.0 a	2.0 b
from AxB/2	xN	100 a	1.6 b	2.0 a	2.6 a	1.6 a
from SNC/3	1N	100 a	1.2 a	3.3 a	3.7 a	1.4 b
from AxB/3	xN	100 a	1.7 b	5.0 a	7.0 b	1.1 a
from SNC/5	1N	100 a	2.1 a	1.7 a	2.8 a	1.9 b
from AxB/5	xN	95.6 a	2.1 a	3.2 b	5.3 b	1.7 a
from SNC/6	1N	100 a	1.6 a	2.1 a	2.9 a	0.9 a
from AxB/6	xN	91.5 a	1.4 a	1.9 a	2.5 a	0.9 a
from SNC (mean)		98.5 -	1.6 -	2.5 -	3.1 -	1.6 -
from AxB (mean)		96.8 -	1.7 -	3.0 -	4.4 -	1.3 -

1N—single-node explant prepared from “SNC” cultures; xN—several node explants prepared from “AxB” cultures; the means marked with various letter are significantly different at α = 0.05. The observations were made on 4 jars (with 8 cultures) for each treatment, except for 6^th^ subculture—5 jars (with 8 cultures) t.

**Table 6 plants-09-01533-t006:** Growth of mountain mulberry ‘Kenmochi”’ in vitro cultures established from one- (1N) and two-node (2N) explants on “SNC” medium.

Explant Type	No of Growing Cultures [%]	Number of Shoots [pcs]	Length of the Longest Shoot [cm]	No of Rooted Shoots [%]	Callus Size [cm]
1N	96.9 a	1.0 a	2.9 a	100 a	0.0 a
2N	100 a	1.0 a	2.5 a	100 a	0.0 a

1N—single-node and 2N—two-node explants prepared from “SNC” cultures; The differences between means marked with the same letter are not significantly proven at α = 0.05; The observations were made on 4 jars (with 8 cultures) for each treatment.

**Table 7 plants-09-01533-t007:** Growth of mountain mulberry ‘Kenmochi”’ in vitro cultures established from shoot-tip (ST) and nodal (xN) explants on standard (AxB) medium.

Explant Type	No of Growing Cultures [%]	Number of Shoots [pcs]	Length of the Longest Shoot [cm]	Total Length of Shoots [cm]	Callus Size [cm]
xN	95.8 a	1.4 a	1.7 a	2.1 a	0.9 a
ST	91.3 a	2.2 b	3.4 b	5.4 b	0.8 a

xN—several node and ST—shoot-tip explants obtained from “AxB” cultures. The means marked with various letter are significantly different at α = 0.05. The observations were made on 4 jars (with 8 cultures) for each treatment.

**Table 8 plants-09-01533-t008:** Chosen traits of acclimated, four weeks old mountain mulberry ‘Kenmochi”’ plantlets of different origin.

Origin of Microshoots	No of Rooted Shoots [%]	No of Shoots [pcs]	Length of the Longest Shoot [cm]	Total Length of Shoots [cm]	Length of Leaf Blade [cm]	Width of Leaf Blade [cm]	Relative Chlorophyll Content [SPAD]
SNC	100 a	1.1 a	6.0 a	6.2 a	3.5 a	2.6 a	23.3 a
AxB	100 a	1.3 a	6.1 a	6.8 a	3.3 a	2.7 a	24.2 a

SNC—shoots prepared from “SNC” cultures, AxB—shoots prepared from “AxB” cultures, forty “AxB” and sixty “SNC” shoots (at least 2 cm long with shoot tip) were used in experiment. The differences between means marked with the same letter are not significantly proven at α = 0.05.

**Table 9 plants-09-01533-t009:** Comparison of two methods of mountain mulberry propagation through in vitro shoot cultures.

Method of Micropropagation	Axillary Branching (“AxB”)	Single-Node Culture (“SNC”)
Cost of media	higher	lower (reduced dose of sucrose and mineral nutrients)
Efficiency of micropropagation		
Initiation of in vitro cultures (explants from young, nursery plants)	easy	easy
Initiation of in vitro cultures (explants from adult, field trees)	possible	possible although hard to get
Number of shoots	significantly higher	significantly lower
Length of shoots	significantly lower	significantly higher
Number of nodes on shoot	similar	similar
Estimated multiplication rate *	lower: 4 (2 × “ST”+ 2 × “xN” explants)	higher: 5 (1 × “ST” + 4 × “1N” explants)
Rooting and acclimation in vivo	similar	similar
Quality of cultures during multiplication		
Growth of callus	too intensive	not observed
Physiological disorders (shoot-tip necrosis, vitrification, chlorosis, *etc*.)	sometimes observed	not observed
Development of adventitious shoots	rather not observed (overgrown callus complicates identification)	not observed
Risk of somaclonal variation	low	very low

*/ estimated on the base of Table 1 (means for 6 subcultures); ST—shoot tip; 1N— single-node; xN—several node explants.

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
