# Peer review of "Micropropagation of Mountain Mulberry (Morus bombycis Koidz.) ‘Kenmochi’ on Cytokinin-Free Medium"

_plants, 2020, doi:10.3390/plants9111533_

Round 1

Reviewer 1 Report

This is a well written article. The applied experimental design, although classic, is developed logically. The results are presented correctly and the discussion is adequate. However, what is missing  in the manuscript are the data on the genetic stability and identity of plants propagated in vitro, under different conditions, compared to intact plants. If the authors have such data, it is good to add them. In addition, it would be good if the reference list is enriched. There is enough data on the subject in the scientific literature.

Specific comments:

  • As far as it is well known that using of the same nutrient media but from different producers led to different responses of plant tissues, it would be a good idea (in the Mat. and Med.) if the sources of media, vitamins and growth regulators is noted.
  • I recommend swapping paragraphs 2.1 and 2.2 and to think about the same in the Results section.

Author Response

Dear Reviewer,

Thank You very much for positive opinion! :)

Unfortunately, we did not perform any genetic analyses :/ We propagated mulberries through axillary shoots which develop from preexisting meristems formed in apex or nodes. In common opinion such method provides maintaining clonal fidelity. Based on my experience it is hard to obtain adventitious shoots on media used in routine micropropagation of mulberries (contrary to Vaccinium cultures). The mulberry explants have to be specially pretreated and placed on other media to obtain adventitious shoots. I have also never found or heard about any off-types among micropropagated mulberry plants in the field. On the other hand we frequently observed that adventitious shoots develop spontaneously in Fragaria and Vaccinium cultures thus they are prone to somaclonal variation. Might be it is a reason, the results of molecular analyses are often presented in articles concerning aforementioned plants and very rare in the case of mulberry micropropagation !?! The two found reports did not pertain plants propagated routinely but obtained from cryopreserved buds or from callus. Nevertheless we added appropriate paragraph statement in Discussion. We enriched also reference list.

We added information about the sources of chemicals in the legend of Table 1.

We also considered swapping aforementioned paragraphs during preparation our manuscript to describe steps of micropropagation in proper order. The problem was that experiment on culture initiation was additional one. Nevertheless we changed the order of description.

Reviewer 2 Report

The research manuscript entitled “Micropropagation of Mountain Mulberry (Morus bombycis Koidz.) ‘Kenmochi’ on Cytokinin-Free Medium”, focus on the comparison of two different methods for micropropagation of mulberry – the classical axillary-branching and the single-node method. The authors present the disadvantages of the commonly used method and propose and explore a new method based on a cytokinin-free medium. The manuscript is well organized, and the experiments, though simple, are sound and logical. The introduction, methods, results and discussion sections are well written and provide enough bibliographic references to support the text. The evidence provided by the authors is sufficient to meet their conclusions. However, some concerns were raised after a careful reading of the manuscript:

  1. the presentation of the figures is not the best and they could be greatly improved. For instance, whenever possible, make the different panels of a given figure the same size and aligned to each other; mark with a letter each panel to better report to each situation presented, both in the text and in the legend; correct the light of some images as some are too bright, while others too dark.
  2. Legends of the figures and tables should be more detailed and give enough information regarding the data in the figure.
  3. The method followed for determination of the chlorophyll content should be described in the methods section.
  4. A graphic or table summarizing the most important parameters quantified, that allowed the authors to reach their conclusions, could be presented in the end to allow the reader a more direct comparison between the two methods.

Author Response

Dear Reviewer,

We are grateful for positive opinion! :)

We hope we improved figures and legends to meet Your requirements. We added information about determination of the chlorophyll content. I think it is good idea to place an additional table summarizing the most important differences between two methods; so I did it. Thank You for such suggestion !

Reviewer 3 Report

Dear Authors,

     I have reviewed this manuscript and I think it is worth to be publish although some points should be improved:

First of all, some methological details should be included for example:

- In Materials and Methods chapter it should be clearly indicated where do the plant material come from?  What about the acessions number? What was the reason for choosing such a variety? Please explain.

- Line 71 – please explain why only one temperature was tested? According to the Authors The temperature was set on 26ºC. It could be interesting what about the regeneration differences between two or three different temperatures.

- In 73-74 lines there is no mentioned about the number of treatments. There is only indicated that „Each treatment was represented by at least 4 jars (x 8 explants)”.

- It is not clear why for single-node culture (‘SN’) and axillary-branching (‘AxB’) two different types of media were tested? It is obvious that usually the tested cultures which were propagated through single node (‘SN’)method should be propagated both on the same medium like control cultures and additionally on modified medium.

- My suggestion is to prepare a new table with number of treatments and types of particular medium in Material and Methods chapter. In this way it will be much easier to visualise the methodology.

- Please write an additional and separate Conclusion chapter.

-  In References chapter there are very few citations. Please add some more citations, for example:

  1. Application of Tissue Culture Techniques for Propagation and Crop Improvement in Mulberry (Morus spp.). K. Vijayan, A. Tikader, J.A. Texeira da Silva
  2. Efficient plant retrieval from alginate-encapsulated vegetative buds of mature mulberry trees. S.K. Pattnaik, Y.Sahoo, P.K. Chand

Author Response

Dear Reviewer,

Thank You for positive review!

Some information about source plant origin was given at the end of manuscript. However now we have added more detailed information in M&M. We set one temperature because we had one growing room. I think we used standard temperature for growing thermophilic plants (mulberries and paulownias). Similar temperatures (25-27oC) were set by many other scientists. As a matter of fact we sometimes kept mulberry cultures in small chambers in 22oC to slow down their growth and 33oC for thermotherapy. The cultures’ reaction and development were similar to one, described for 26oC. The shoot elongation was slightly stimulated (but callus growth also) in 33oC. However, we did not carry out strict experiment and measurements. It is only personal observation. It was not documented. Thus we cannot present it in the manuscript.

We added the table with media composition in M&M and gave more precise data about the number of cultures tested in experimental subcultures under the tables.

We separated conclusion chapter and added new table contained main differences between two ways of micropropagation.

Hm, I do not understand that remark as the results of such experiments were presented in Table 3? Might be our descriptions were not sufficiently clear :/ On the other hand, the Reviewer 4 has written that such experiments (reciprocal switching from ‘SN’ to ‘AxB’ method) are not consistent with the research idea (exclusive application of cytokinin-free medium). We have already explained in Discussion (in M&M also) that that establishment of strict experiments was not possible because of the different morphology of shoots obtained through compared methods and difficulty of application and preparation the same explants. Single-node explants prepared from cultures grown on BA-supplemented medium (AxB) are short (sometimes 2 mm) whereas from ‘SN’ cultures are long (sometimes more than 10mm). We added additional picture to visualise the differences between shoots obtained from ‘AxB’ and ‘SN’ cultures. It worth mentioning, that in our previous observations the very short single node explants placed on BA-containing medium produce often big callus and even do not develop new axillary shoots.

We have read and included suggested articles in the reference list, and other ones, as well.

Kind regards, WL

Reviewer 4 Report

Dear Editor and Authors,

The manuscript concerns a question of micropropagation of mulberry (Morus sp.), one of the economically important trees grown in Asian countries (although there are some micropropagation protocols for Morus sp. available).

I find the manuscript suitable for publication after major revision.

Questions to be addressed:

  1. reference update – the problem, experiment and discussion should be supported by the possibly new publications that review the current state of the research field. The Authors cited 16 publications and 10 of them come from the previous century;
  2. *the introduction and the problem to be solved seem a bit unclear – On the one hand one may read: “It seems that mulberry in vitro cultures synthesize endogenous auxins as shoot cultures spontaneously roots during proliferation stage…”(line 42) and “As an apical dominance is observed it is worth to check whether mulberries could be propagated in vitro without application of plant growth regulators according to single-node culture method on medium devoid of cytokinins” (line 45). On the other hand– IBA was introduced to the tested medium. What was the purpose of adding IBA?

*“In supplementary experiments, to check whether the ‘switch’ from SN to AxB method is possible, the explants prepared from SN cultures were grown on control ‘AxB’ medium” (line 75). What was the purpose of testing of the “switching possibility”? Wasn’t it the BA presence (and subsequent callus induction) that the Authors were trying to avoid?

Please, justify the method/medium you have tested to clarify the strategy of the experiment (also to keep the introduction comprehensible to scientists working outside the topic of the MS).

  1. results/discussion – is there a significant relationship observed between a number of nodes per explant and a number of shoots per explant? If yes, what would be the Authors’ evaluation of the compared methods?

Author Response

Dear Reviewer,

Thank You for positive recommendation and accurate comments !

I read new articles and expanded reference list according to requests of 3 Reviewers.

I added also short description of two studied method for scientists working outside the topic of the TC.

In my old experiments we found that shoot tip explants (of 4 mulberry clones) root better on medium devoid of auxins [Litwińczuk et al 1999, in polish :/]. However, rooting of nodal explants was worsened. As in the present study we used single-node explants we added small dose of auxin to solve that problem. On the other hand we’d like to underline, that we used IBA in low concentration (0.05 mg/L). Other scientists usually apply auxins (IBA, NAA, IAA, 2,4-D) in higher doses; at least (0.1-1.0 mg/L) even till 5 mg/L.

Truly, the small experiment on “switching possibility” looks like “art for art being” !? I admit we established it just for curiosity to check whether previous treatment may positively influence behavior of cultures on other medium. In articles devoted to mulberry cultures there are many examples that pretreatment on explants influences organogenesis of adventitious shoots, bud dormancy, etc. We found adverse after-effect of media switching. Thus we obtained additional evidence such 'switch' is not recommended. We hope You will accept leaving description of this experiment.

No, I did not found clear, positive relationship between a number of nodes per explant and a number of shoots per explant. Usually ST (with similar or lower number of nodes) produces more shoots than xN explant on ‘AxB’ medium. Such information was given in tables 6 and 7. By the way – I found and corrected evident mistakes in Table 6. Unfortunately we did not count the number of nodes on shoots in most performed experiments (only for the first subculture). Fortunately we still kept cultures growing on both media. I did additional measurement and took pictures. Surprisingly for me both cultures did not differ significantly in relation to the number of nodes per shoot. Details are given in Discussion.

Kind regards, WL

Round 2

Reviewer 4 Report

Dear Authors,

thank you for all the correction you have made.

I my opinion the manuscript could be published in the present form.

Best regards,

Reviewer